# ABC transporter functions as a pacemaker for sequestration of plant glucosides in leaf beetles

**Anja S Strauss[1]\*, Sven Peters[2], Wilhelm Boland[1], Antje Burse[1]\***

[1]Department of Bioorganic Chemistry, Max Planck Institute for Chemical Ecology, Jena, Germany; [2]Department of Ophthalmology, University Hospital Jena, Jena, Germany

**Abstract** Plant-herbivore interactions dominate the planet's terrestrial ecology. When it comes to host–plant specialization, insects are among the most versatile evolutionary innovators, able to disarm multiple chemical plant defenses. Sequestration is a widespread strategy to detoxify noxious metabolites, frequently for the insect's own benefit against predation. In this study, we describe the broad-spectrum ATP-binding cassette transporter *Cp*MRP of the poplar leaf beetle, *Chrysomela populi* as the first candidate involved in the sequestration of phytochemicals in insects. *Cp*MRP acts in the defensive glands of the larvae as a pacemaker for the irreversible shuttling of pre-selected metabolites from the hemolymph into defensive secretions. Silencing *Cp*MRP in vivo creates a defenseless phenotype, indicating its role in the secretion process is crucial. In the defensive glands of related leaf beetle species, we identified sequences similar to *Cp*MRP and assume therefore that exocrine gland-based defensive strategies, evolved by these insects to repel their enemies, rely on ABC transporters as a key element.

**\*For correspondence:** astrauss@ice.mpg.de (ASS); aburse@ice.mpg.de (AB)

**Competing interests:** The authors declare that no competing interests exist.

**Reviewing editor**: Marcel Dicke, Wageningen University, The Netherlands

## Introduction

For millions of years, insects have relied on plants as a food source. To impede herbivory, plants have developed several morphological and biochemical traits; one of those is based on toxic secondary metabolite production. Insects, in turn, have evolved ingenious detoxification strategies, including the process of sequestration, to overcome the chemical plant defenses (*Sorensen and Dearing, 2006*; *Li et al., 2007*; *Opitz and Muller, 2009*; *Boeckler et al., 2011*; *Winde and Wittstock, 2011*; *Dobler et al., 2012*). These counter-mechanisms thereby affect the ecology and evolution of plants (*Ehrlich and Raven, 1964*; *Agrawal et al., 2012*; *Hare, 2012*). The phenomenon of sequestration involves the uptake, transfer, and concentration of occasionally modified phytochemicals into the hemolymph, cuticle, specialized tissues or glands. Numerous species from almost all insect orders have evolved the ability to sequester chemicals (*Duffey, 1980*; *Nishida, 2002*; *Opitz and Muller, 2009*). Frequently the sequestered toxins are used by insects for their own defense, as is the case in leaf beetles (Chrysomelidae) (*Meinwald et al., 1977*; *Pasteels et al., 1990*; *Gillespie et al., 2003*). Up to now, the most comprehensive knowledge of sequestration processes has been obtained from juveniles of the leaf beetles belonging to the taxon Chrysomelina (*Soetens et al., 1998*; *Termonia et al., 2001*). The chemical defenses of these larvae are made up of compounds that are either sequestered from their host plants or synthesized de novo. Regardless of their origin, these compounds are transferred into nine pairs of specialized exocrine glands that are found on the back of the larvae (*Hinton, 1951*; *Pasteels and Rowell-Rahier, 1991*; *Pasteels, 1993*). According to morphological studies, each defensive gland is composed of a number of enlarged secretory cells, which are in turn connected to a chitin-coated reservoir. The secretory cells are always accompanied by two canal cells that

**eLife digest** For millions of years, plant feeding insects have been locked in an arms race with the plants they consume. Plants have evolved defensive strategies such as the ability to produce noxious chemicals that deter insects, while many insects have evolved the means to thwart this defense and even turn it to their own advantage. The larvae of the poplar leaf beetle, *Chrysomela populi,* sequester toxic plant compounds in specialized glands on their backs and use these compounds to defend themselves against predators. The glands are lined with chemically inert chitin, the substance that makes up the insect exoskeleton, and the deterrent chemicals are released whenever the insect is threatened.

Now, Strauss et al. have identified a key transport protein used by the larvae to move toxic plant compounds to these glands. This transport protein belongs to a family of membrane proteins called ABC transporters, which help to shuttle substances out of cells or into cell organelles using energy produced by the hydrolysis of ATP molecules. The gene for this transporter is expressed in the glands of the leaf beetles at levels 7,000 times higher than elsewhere in the larvae.

Larvae that lack a functional version of the transporter gene continue to grow, but are unable to defend themselves against predators. Similar genes are found in other species of leaf beetle, suggesting that this type of transporter has been retained throughout evolution. Moreover, the transporter is not specific to a particular plant toxin; this enables leaf beetles to eat many different types of plants and boosts their chances of survival should a previous food source disappear.

form a cuticular canal, which connects the secretory cell with the reservoir (***Noirot and Quennedy, 1974***). When disturbed, the juvenile beetles evert their glandular reservoirs and present droplets of secretions.

In Chrysomelina larvae, all compounds reaching the glandular reservoir via the hemolymph are glucosides that are converted enzymatically into the biologically active form within the reservoir (***Pasteels et al., 1990***). Thus, the glands also secrete enzymes for the final metabolic conversion of precursors into defensive compounds in the reservoir. The ability to sequester plant glucosides is considered an energy-saving, monotypic adaptation within Chrysomelina (***Figure 1B***), given the phylogenetic evidence that this process evolved from an ancestral autogenous biosynthesis of deterrent monoterpenes (iridoids) (***Termonia et al., 2001***).

The poplar leaf beetle *Chrysomela populi*, is an example of an obligate-sequestering species, and its larvae incorporate the phenolglucoside salicin from the leaves of their salicaceaous food plants (***Pasteels et al., 1983***; ***Kuhn et al., 2004***). In the reservoir of their defensive glands, the salicin is then metabolized into the volatile deterrent salicylaldehyde (***Michalski et al., 2008***). Additionally, in *Chrysomela lapponica* several glucosidically bound alcohols are simultaneously imported, resulting in a diversity of compounds, especially of esters, in the exudate of the larvae (***Hilker and Schulz, 1994***; ***Schulz et al., 1997***; ***Kirsch et al., 2011***; ***Tolzin-Banasch et al., 2011***). Physiological studies on de novo iridoid-producing, salicin sequestering, and ester-producing larvae using thioglucosides have indicated a complex influx–efflux transport network that guides the plant-derived glucosides through the insect body (***Discher et al., 2009***; ***Kuhn et al., 2004***). Presumed intestinal carriers in the gut epithelial cells allow a broad spectrum of secondary metabolites to enter the hemolymph, a process that is accompanied by a similar non-selective excretion via the Malpighian tubules. Furthermore, thioglucosides are being selectively accumulated, up to 500-fold, into the reservoir from a hemolymph pool, suggesting an active transport system is at work (***Feld et al., 2001***; ***Kuhn et al., 2004***). By employing the obligate salicin sequestering species *C. populi*, we focus on deciphering the transport processes involved in the sequestration of glucosides in the defensive glands of chrysomelid larvae.

The active ATP binding cassette (ABC) transporters are well-known key-components of various detoxification mechanisms in all phyla of life (***Sipos and Kuchler, 2006***; ***Leprohon et al., 2011***; ***Holland, 2011***; ***Broehan et al., 2013***). In eukaryotes, they translocate a wide variety of compounds from the cytoplasm to the extracellular space or to intracellular compartments. Their role in the sequestration of plant secondary metabolites in insect herbivores, however, has not yet been investigated (***Karnaky et al., 2000***; ***Sorensen and Dearing, 2006***).

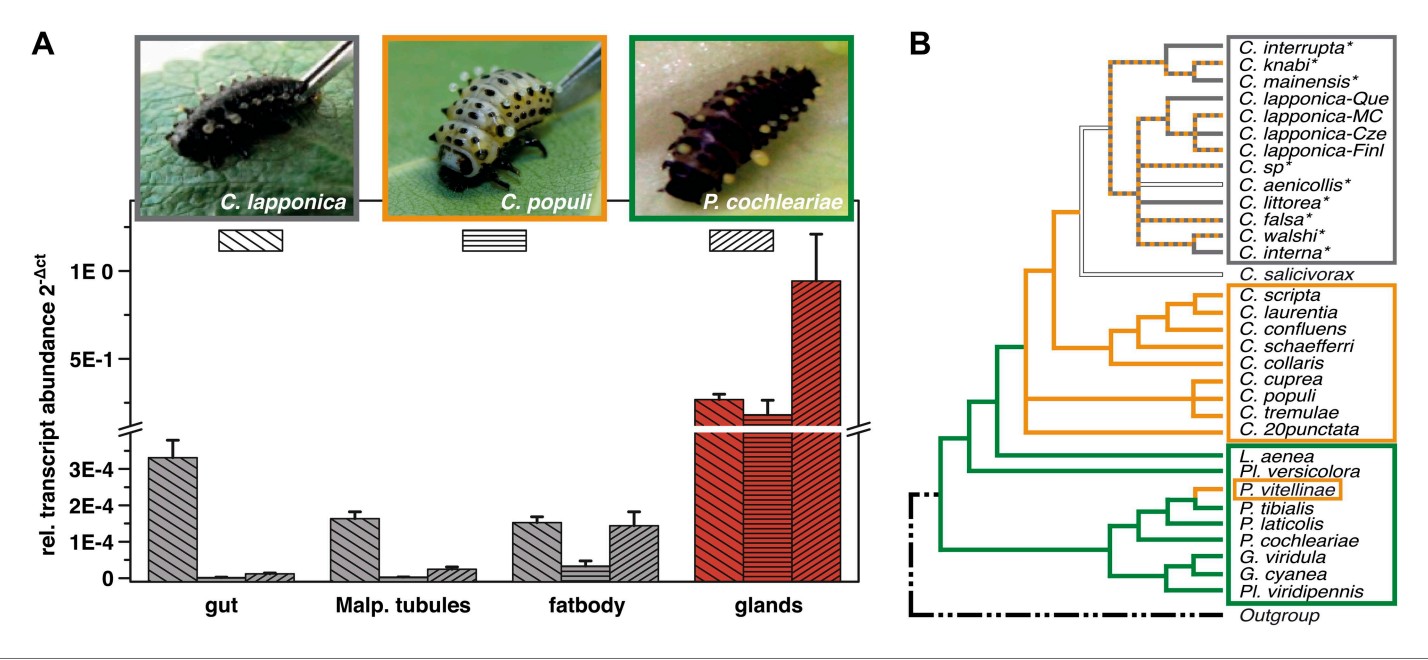

**Figure 1**. Glandular tissue-specific transcript level of *cpmrp* and its homologous sequences. (**A**) Relative transcript abundance of *cpmrp* (*C. populi*) and its homologous sequences from *C. lapponica* and *P. cochleariae* in different larval tissues (n = 3–4, mean ± SD) assigned to (**B**) their phylogenetic group and chemical defense strategies based on maximum parsimony reconstruction (according to *Termonia et al., 2001*). Green, autogenous group of monoterpene iridoid producers; orange, obligate-sequestering group; gray, *interrupta* group with mixed metabolism that evolved the biosynthesis of butyrate-esters.

Here we identify *Cp*MRP as a class C-ABC transporter in the defensive glands of *C. populi*. We demonstrate *Cp*MRPs transport activity for plant-derived glucoside precursors. In the absence of *Cp*MRP, larvae of *C. populi* develop normally but lack defensive secretions that assign a key role for *Cp*MRP in the process of sequestration of salicin. We also describe a sequestration model in which ABC transporters play a key role and discuss their general relevance in exocrine glands of Chrysomelina species.

## Results and discussion

Screening of expression levels of transcript sequences encoding ABC transporter motifs revealed a putative candidate, referred to here as *Cp*MRP. It displayed an exceptionally high transcript level in the glandular tissue, exceeding that in the gut and Malpighian tubules by more than 7000-fold (*Figure 1A*).

Among all known and functionally characterized ABC transporters, the deduced amino acid sequence of *cpmrp*, which contained 1331 residues (154.9 kDa), shares the highest sequence similarity of 61% (41% sequence identity) to the human homologous multidrug resistance-associated protein MRP4 (ABC subfamily C) (*Lee et al., 1998*). The predicted protein of *Cp*MRP from *C. populi* possesses the typical structural elements of ABC transporters (*Zolnerciks et al., 2011*); these consist of four domains: two TMDs (transmembrane domains), harboring six proposed transmembrane spans and two NBDs (nucleotide-binding domains), containing Walker A and B boxes (sequences GPVGAGKS and VYLMD, respectively), separated by an ABC signature motif (sequence LSGGQRARINLARAI). Additionally, we conducted a 3D structure modeling of *Cp*MRP (*Figure 2D*) to support the conclusions of the sequence alignment and to illustrate the localization of characteristic sequence motifs. Both sequence alignment and structure modeling suggest that the newly identified protein *Cp*MRP is very likely an ABC transporter.

In the ancestral de novo iridoid-producing species *Phaedon cochleariae*, we have identified a sequence with 86% amino acid identity to *Cp*MRP and in the more derived species *C. lapponica* we identified a homolog of *Cp*MRP sharing 93.7% amino acid identity. Both the sequences share the same

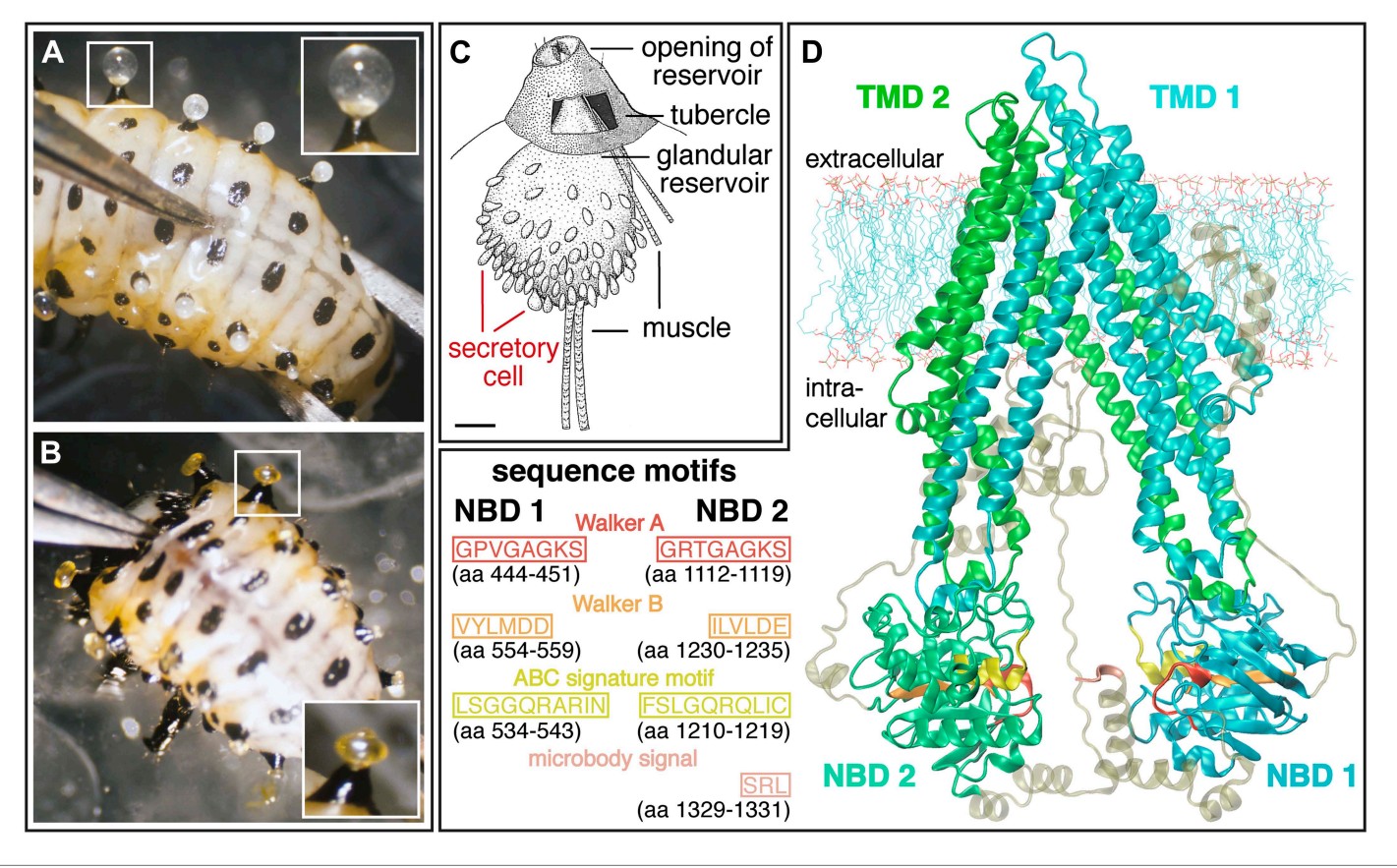

**Figure 2**. Silencing effect and 3D-structure model of *Cp*MRP. (**A** and **B**) Production of defensive secretions is disrupted in *Cp*MRP knockdown L3 larvae (**B**) compared to the phenotype of the control larvae (**A**). (**C**) Drawing of dissected glandular tissue of *C. populi* according to **Hinton, 1951** with relaxed reservoir in contrast to the everted reservoir in insets of (**A**) and (**B**). (**D**) 3D-model of *Cp*MRP, embedded in a lipid bilayer, illustrating its probable correct global topology based on I-TASSER (TM-score of 0.52 ± 0.15; C-score: −1,57) and the localization of characteristic sequence motifs.

transcription pattern like *Cp*MRP, being highly expressed in the larval glands only (**Figure 1A**). Altogether this suggests that there is a highly conserved ABC transporter among Chrysomelina species that has an important ecological role. Given the uniform architecture and morphology of the defensive system (**Hinton, 1951**; **Noirot and Quennedy, 1974**) in Chrysomelina larvae, we expect functional similarities on the molecular level. We focus in our study on *Cp*MRP as representative of an obligate-sequestering species among Chrysomelina.

To verify whether the transcript abundance of *cpmrp* is also reflected in the protein level of the defensive glands, we carried out immunohistochemical localization. The staining of full-body sections from juvenile *C. populi* showed that *Cp*MRP was exclusively localized in the defensive glands (**Figure 3—figure supplement 1**). In more detail, *Cp*MRP was present neither in the canal-forming cells (**Figure 3A—C1, C2**) nor in the canal itself (**Figure 3Bb—Cc**) but, rather in the secretory cells (**Figure 3A**). Intriguingly, mapping *Cp*MRP at subcellular magnification provided evidence for the exclusive localization within the secretory cells attached to the reservoir. The intracellular distribution of *Cp*MRP resembles a hollow sphere with a distinct reticular pattern (arrows in **Figure 3C**, **Figure 3E**, **Videos 1 and 2**). Its intracellular presence in vesicular and reticular structures (**Figure 3Bc**) was corroborated by its co-localization with Bodipy-stained intracellular membranes (**Figure 3C,D**). Starting at the outside of the secretory cell, *Cp*MRP was not detected in the basal lamina or in the adjacent basal infoldings (~5 µm) (indicated with Bi in **Figure 3Ba,C**). Instead, directly after the basal infoldings, we observed a sharp transition to a spherical zone of about 2–5 µm; here, we noted

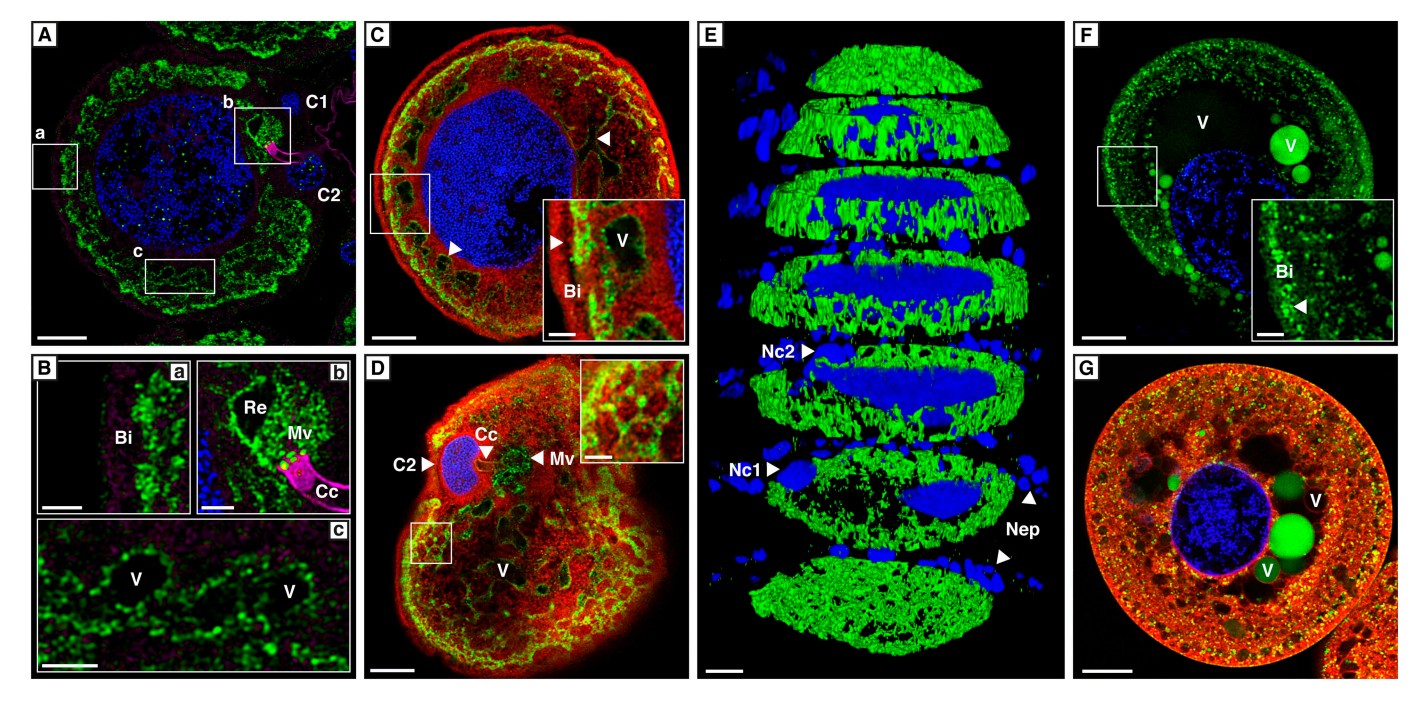

**Figure 3**. Localization of *Cp*MRP in secretory cells of *C. populi*. (**A–G**) High-magnification optical sections through secretory cells. (**A–E**) Immunohistochemical staining of *Cp*MRP (green) in fixed secretory cells. *Cp*MRP staining was confined to intracellular Bodipy-stained membrane structures and displayed a distinct reticular pattern. (**B**) Extracted cutouts of (**A** and **C**) optical section through the nucleus, (**D**) optical section above the nucleus of the secretory cell, (**E**) 3D stack displaying *Cp*MRPs primarily spherical distribution. (**F** and **G**) CDCFDA staining for vacuolar esterase activity (green) in live cells. A multitude of vacuoles are present that vary in their enzyme content. Bi = basal infoldings, C1 and C2 = canal cells, Cc = cuticular canal, Mv = microvilli, Nc1, Nc2 = nucleus of canal cells, Nep = nuclei of epithelium cells, Re = extracellular room, V = vacuole, Blue, nuclear staining; Red, Bodipy-stained intracellular membrane; Magenta, false color-coded autofluorescence. Scale bars, 20 μm or 5 μm (insets).

The following figure supplements are available for figure 3:

**Figure supplement 1**. Localization of *Cp*MRP in whole larvae cryosections of *C. populi*.

the most dense *Cp*MRP presence within the entire cell (***Figure 3A–C*** [inset arrow] and ***Figure 2D***). Moreover, according to vacuolar esterase activity, demonstrated by CDCFDA staining (***Pringle et al., 1989***), the subcellular localization of *Cp*MRP (***Figure 3F,G*** [inset arrow]) correlates with cellular storage compartments (***Figure 3A–E***).

Employing RNA interference (RNAi) to verify *Cp*MRP's relevance for salicin sequestration in vivo, we were able to demonstrate its key role in the secretion of defensive compounds. By comparing developmental traits among *C. populi* larvae after injecting *cpmrp*-dsRNA or *gfp*-dsRNA as a control, we found that silencing *cpmrp* had no influence on larval growth (***Figure 4A***). However, about 10 days post-injection, the *cpmrp* knockdown larvae completely lost their ability to respond to stimulation with droplets of defensive secretion (***Figures 4B and 2B***). The secretions began to diminish at day 8. On the basis of transcript abundance, *cpmrp* mRNA was reduced to a basal level of 15–20% within 2–3 days and persisted until the larvae pupated (***Figure 4C***). ***Figure 4D*** summarizes the immunohistochemical analysis of *Cp*MRP expression in the secretory cells. At day 3 post-injection, both the *cpmrp* knocked-down and control secretory cells displayed a similar pattern of *Cp*MRP distribution (***Figure 4Da,b***). At later time points, however, the *Cp*MRP expression in the RNAi group was strongly reduced in comparison to the *gfp* control (***Figure 4Dc—f***), which is in agreement with Western blot analysis (***Figure 4—figure supplement 1***). *Cp*MRP decayed exponentially to a relatively low basal level with a half-life of about 1 day, suggesting a degradation of *Cp*MRP that is linearly proportional to its concentration (***Figure 4—figure supplement 2***).

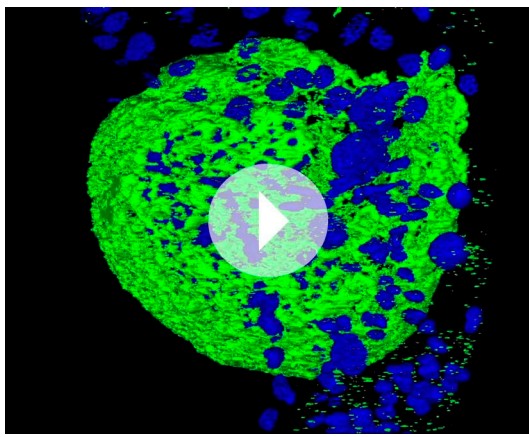

**Video 1**. 3D representation of *Cp*MRP localization within a secretory cell. Exterior view of a secretory cell of *C. populi* based on immunohistochemical staining (green, *Cp*MRP; blue, nuclei stain). The z-stack was acquired with a resolution of x = 0.146 μm; y = 0.146 μm and z = 0.500 μm. The smallest dimension of the depicted secretory cell is about 100 μm.

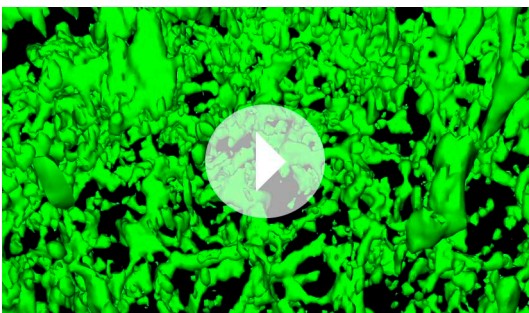

**Video 2**. 3D representation of *Cp*MRP localization within a secretory cell. Interior view of **Video 1**. The camera is centrally positioned within the nucleus and rotates within the plane of the first frame of **Video 1**, initially pointing to the area between the nuclei of the two canal cells.

This proportionality has already been reported in human hepatocytes (*Pereg et al., 2010*; *Popov et al., 2010*; *Nakagawa et al., 2011*). Compared to a half-life of 5 days for the ABC transporters MDR1 and MDR2 reported in hepatocytes (*Kipp and Arias, 2002*), the rate of *Cp*MRP turnover seems relatively high. Qualitative microscopic observations showed that secretory cells tended to increase the size of storage compartments, presumably vacuoles, of the *Cp*MRP knockdown larvae.

Our transport studies in *Xenopus laevis* oocytes revealed that *Cp*MRP is a transporter for salicin (*Figure 5A*), the naturally sequestered host–plant precursor of *C. populi*. In order to test the selectivity of *Cp*MRP, we chose a mixture of glucosides among plant precursors and non-precursor glucosides for comparative transport assays (*Figure 5C,D*). We applied an equimolar mixture of salicin (1), 8-hydroxygeraniol-*O*-glucoside (2), the early precursor of the iridoid monoterpene pathway found in *P. cochleariae* and phenylethyl-*S*-glucoside (3) that represents a substrate mimicking an *O*-glucoside sequestered by *C. lapponica*. This mixture was tested in feeding and hemolymph injection experiments on *C. populi* and revealed the specific transport of salicin to the reservoir (*Discher et al., 2009*). However, *Cp*MRP did not discriminate significantly between the substrates. For phenylethyl-*S*-glucoside and thiosalicin (6) the transport activity of *Cp*MRP was slightly reduced compared to salicin (*Figure 5C,D*). Moreover, the sugar moiety of the substrates (comparing salicin and its galactoside analogue (4) (*Figure 5D*), further significantly lowered the transport activity of *Cp*MRP, which is consistent with previous data obtained by feeding experiments (*Kuhn et al., 2004*). Based on the apparent Km of 5.8 (mM) for salicin, *Cp*MRP (*Figure 5B*) seems to be a low-affinity transporter. From our comparative transport assays we assume that *Cp*MRP functions as low-affinity glucoside transporter with a broad glucoside spectrum.

Taken together, these data support a sequestration model inside the secretory cells of *C. populi* in which *Cp*MRP plays a key role as a pacemaker (*Figure 6*). We assume that within the described zone of highest *Cp*MRP density (*Figure 3A,B,D*), salicin is trapped in storage vesicles as soon as it enters the secretory cells. The constant vesicular accumulation of plant glucoside precursors and further irreversible translocation into the reservoir keeps the glucoside concentration low inside the secretory cell. By this fact, we suggest a first filter for specific glucosides (salicin, in the case of *C. populi*) at the hemolymph-exposed plasma membrane of the secretory cell, which might depend on a gradient-driven, energy-independent transporter. Accordingly, *Cp*MRP does not just dictate the transport rate of this transporter in the plasma membrane, rather, it determines the effectiveness and energy coupling of the entire sequestration process as a pacemaker.

*Figure 6* further illustrates the fate of storage compartments. The apical part facing the lumen of the gland is a brush border membrane where storage vesicles are secreted via exocytosis

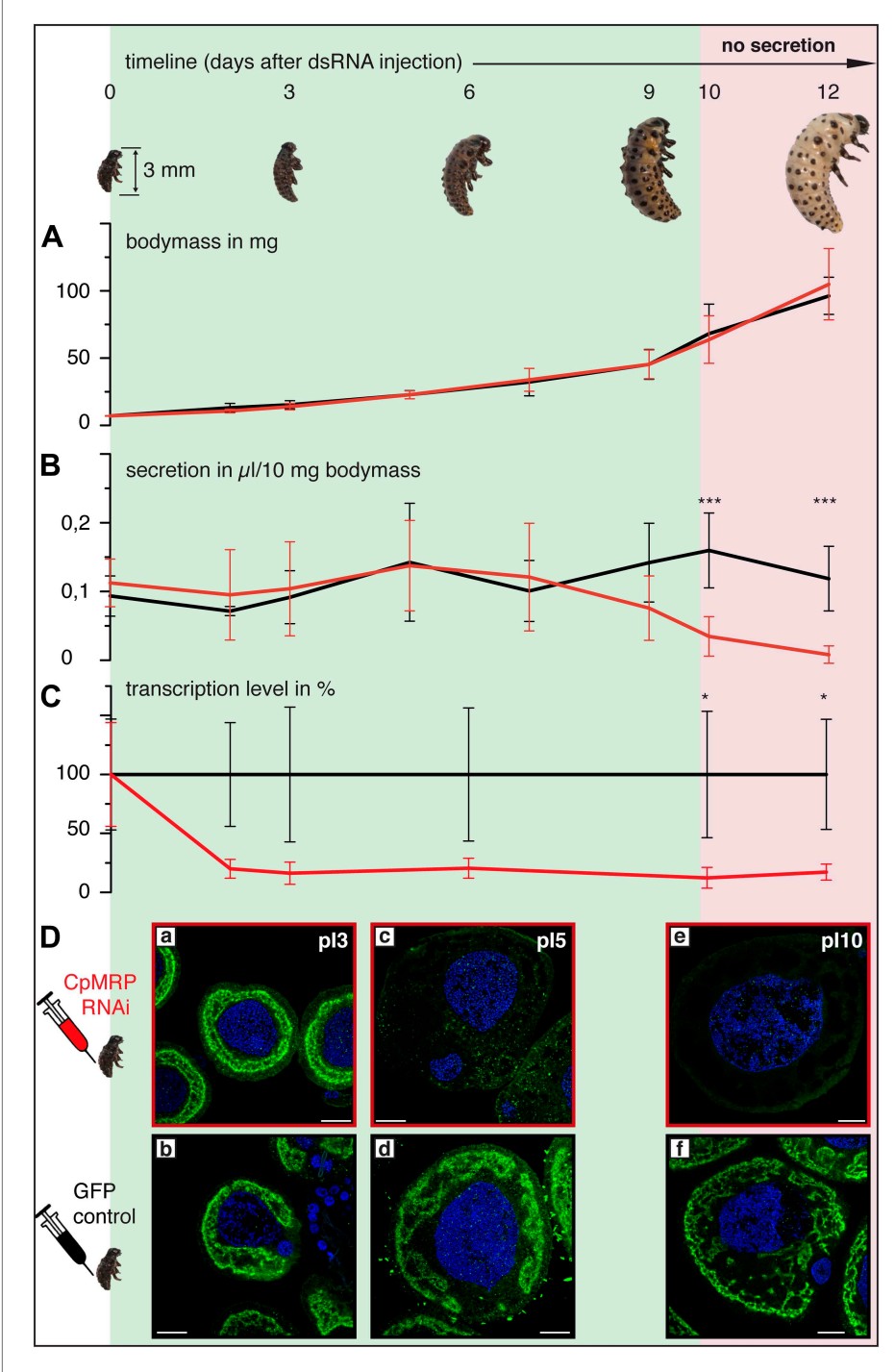

**Figure 4**. Timeline of different *Cp*MRP knockdown effects. (**A**–**D**) *C. populi* larvae development following *Cp*MRP knockdown by dsRNA injection into the larval hemocoel (d 0). *Cp*MRP knockdown effects; red/(**D a, c, e**) were compared to the *gfp*-injected control larvae; black/(**D b, d, f**) at different developmental stages. (**A**) Larval fitness (body mass) was not influenced by *Cp*MRP knockdown (n > 10, mean ± SD). (**B**) *Cp*MRP knockdown larvae lack defensive secretions 10 days after dsRNA injection (n > 10, mean ± SD). (**C**) Transcriptional level of *cpmrp* inside the glands (each time point contains n = 3 (biological replicates), mean ± SD). (**D**) *Cp*MRP protein level decreased after dsRNA injection—Green, *Cp*MRP; Blue, nuclear stain. Scale bars, 20 µm; plx = x days post dsRNA-injection. Asterisks represent significant differences in *cpmrp*-silenced larvae compared to *gfp*-injected control larvae (*p≤0.05, ***p≤0.001).
*Figure 4. Continued on next page*

*Figure 4. Continued*

The following figure supplements are available for figure 4:

**Figure supplement 1**. Effects of *cpmrp* silencing on the protein level of the glandular tissue of *C. populi*.

**Figure supplement 2**. Degradation kinetics of *Cp*MRP in secretory cells of *C. populi*.

**Figure supplement 3**. DNA alignment of *cpmrp*–related ABC transporter sequences in *C. populi*.

**Figure supplement 4**. Evaluation of possible off-target effects of *Cp*MRP dsRNA in larval tissue of *C. populi*.

(*Noirot and Quennedy, 1974*). It is invaginated and forms an extracellular room that is connected to the reservoir by the canal. The well-known structure of exocrine glands gives hints at exocytic processes on the basis of a microvilli membrane. That *Cp*MRP is present in the microvilli membrane (*Figure 3Bc*), indicates that this is where the exocytosis of *Cp*MRP-containing storage vesicles takes place. After

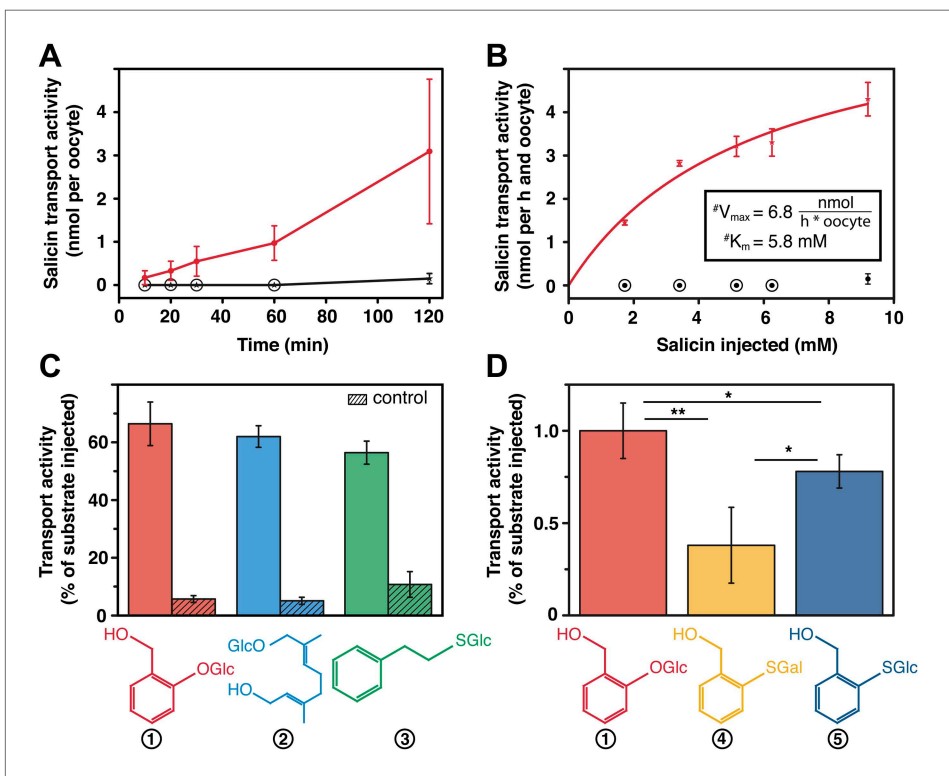

**Figure 5**. Salicin transport activity of *Cp*MRP in *Xenopus laevis* oocytes. (**A–D**) Transport activity was determined by quantifying the substrate efflux in the oocyte incubation medium of *cpmrp*-RNA compared to water-injected control oocytes via HPLC-MS. (**A**) Time course of *Cp*MRP-dependent salicin efflux after the injection of 5 nmol salicin (incubation time: 1 hr, n = 5, mean ± SD). Red, *Cp*MRP-expressing oocytes; Black, water-injected control. (**B**) Concentration dependence of *Cp*MRP-mediated salicin transport (red); water-injected control in black (n = 5, mean ± SD, #: apparent, encircled data point: not detectable). (**C**) Comparative transport assays of *Cp*MRP activity with a substrate mixture (salicin (1), 8-hydroxygeraniol-*O*-glucoside (2) and phenylethyl-*S*-glucoside (3)). Open bar, transport activity of *Cp*MRP-expressing oocytes. Crosshatched bar, transport activity water-injected control oocytes. (**D**) Comparative transport assays of *Cp*MRP salicin transport activity to thiosalicin (5) and its galactoside analogue (4) and thiosalicin (incubation time: 1 hr, n = 10, mean ± SD). Asterisks represent significant differences among indicated substrates (*p≤0.05, **p≤0.01). Encircled data points represent undetectable concentrations.

excreting the glucoside, *Cp*MRP is most likely recycled to recover the protein in the dense packaging zone of storage vesicles/vacuoles.

## Conclusion

In the present study, we show that *Cp*MRP is required to maintain defensive secretion in *C. populi*. Our results demonstrate that *cpmrp*-silenced larvae are defenseless because they lack defensive secretions. Functionally, *Cp*MRP is a transporter for plant derived glucoside precursors present in storage compartments as well as in the microvilli membrane of the secretory cell. Therefore, our results have led us to propose a functional model of sequestration based on *Cp*MRP as the key element (*Figure 6*). The identification of transporter sequences highly similar to *Cp*MRP in the larval glands of other Chrysomelina species (*P. cochleariae* and *C. lapponica*) strongly implies that broad-spectrum ABC transporters involved in the sequestration of plant-derived metabolites are commonly present in the defense mechanism among Chrysomelina.

These results, together with our published data, lead us to conclude that the sequestration of plant glucosides in Chrysomelina larvae is the result of the presence of several barriers with various degrees of selectivity: (1) those controlling the non-selective uptake of plant-derived glucosides from the gut lumen into the hemolymph and their excretion by the Malpighian tubules (together these barriers are relevant for nutrition), (2) those controlling the selective transfer from the hemolymph into the secretory cells and (3) those controlling the secretion into the reservoir where the broad-spectrum ABC transporter acts as a pacemaker. This functional arrangement of a non-selective and a selective transporter in the defensive system seems to be common to many different leaf beetles (*Discher et al., 2009*). This peculiar import system also facilitates the occasional host plant shifts of leaf beetles caused by parasite pressure (*Agosta et al., 2010*). After the shift to a new host plant only the selective transport element needs to adjust to the new metabolites; all other transport elements may remain unchanged due to their broad substrate tolerance. This assumption is supported by the observation of *Cp*MRP homologs in different leaf beetles and beyond (*Tribolium Genome Sequencing Consortium, 2008*); however, none of them has been functionally characterized or localized as yet. The identification of *Cp*MRP as a non-selective pacemaker involved in the sequestration of plant-derived glucosides highlights how insects counter plant chemical defenses to evolve new functions for the plant-derived toxins as allomones.

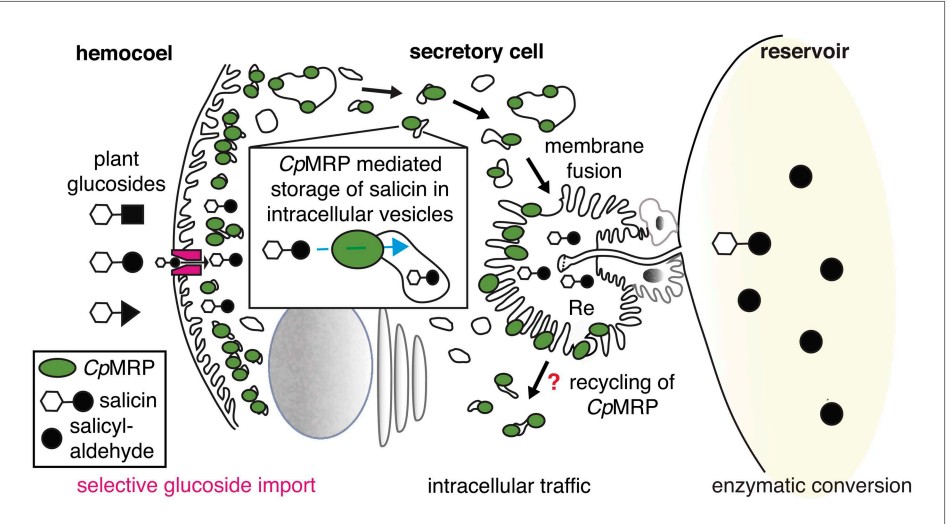

**Figure 6**. *Cp*MRPs pacemaker function and sequestration model. Schematic view of our sequestration model through a secretory cell (see *Figure 2C*: overview of the defensive system, secretory cells are indicated in red). Different plant glucosides (black triangles, circles and squares joined to a glucose molecule indicated by a white hexagon) circulate in the hemocoel. *Cp*MRP dictates (pacemaker function) the transport rate of a still unknown selective, maybe gradient-driven transporter (magenta) for salicin in the plasma membrane by a constant accumulation of salicin in intracellular vesicles. These vesicles are tracked via exocytosis to the reservoir where the enzymatic conversion of salicin to salicylaldehyde takes place.

**Table 1.** Oligonucleotide primers

| Gene name | Primer name | *cpmrp* RACE |
| --- | --- | --- |
| *cpmrp Gene-Bank: KC112554* | 3´RACE | GCACGGTCCTGACTATAGCGCACAGGC |
| | 5´RACE | CCTGCCCCCGTTCTTCCCACAATACC |
| | 2nd 5´RACE | GGTGGAGGCCTGCATGGTCAGCTTGC |
| | 5´nested | CGGCGTCTCGAATGGACCTTCCGTGTCG |
| | 3´nested | GGAGAGATGGTGGGAGTATGACCACCCC |

| Gene name | Primer name | Primer for ds RNA generation |
| --- | --- | --- |
| *cpmrp Gene-Bank: KC112554* | fwd | GATTAATACGACTCACTATAGGCGACTAAGTGTGAACTAGTCGGTGC |
| | rev | GATTAATACGACTCACTATAGGGAGACTTGTCTCCACAGCAGATAG |
| *gfp UniProtKB:P42212.1* | fwd | TAATACGACTCACTATAGGGAGATGGCTAGTAAGGGA |
| | rev | TAATACGACTCACTATAGGGAGATTATTTGTAGAGTTC |

| Gene name | Primer name | qPCR Primer |
| --- | --- | --- |
| *cpRP-L45 GeneBank: JX 122918* | fwd | CACTGGAATCCAAAGTGGAAACTG |
| | rev | CTGCCTTTCAACCCATGGTC |
| *cpActin GeneBank: JX122919* | fwd | ACGTGGACATCAGGAAGGAC |
| | rev | ACATCTGCTGGAAGGTGGAC |
| *pcRP-L8 Gene-Bank: JX122920* | fwd | CATGCCTGAAGGTACTATAGTGTG |
| | rev | GCAATGACAGTGGCATAGTTACC |
| *cpmrp Gene-Bank: KC112554* | fwd | CCTGGATCCATTCGATGAGT |
| | rev | AGTATCGCCCTCGCTAGACA |
| *pcmrp Gene-Bank: KF278996* | fwd | CTCTAGACATCATGGTCACAGA |
| | rev | GGCATATCAACTGTCGTTGTC |
| *clapmrp Gene-Bank: KF278997* | fwd | CCATCTGGCAAATTTGAAGATTTC |
| | rev | AGTATTGCCCTCGCTAGACA |
| *off-target_OT-1* | fwd | GGAATTCGAGGACCAGTTGC |
| | rev | GGCATATCAACTGTCGTTGTC |
| *off-target_OT-2* | fwd | CCATCTGGCAAATTTGAAGATTTC |
| | rev | CAGTAACTACGAAGTCTAGAGAG |
| *off-target_OT-3* | fwd | GAATATCAGATGCCGACCTG |
| | rev | GGATGGCTCTGGCAAGG |
| *off-target_OT-4* | fwd | GACGACGTGCTTTATAGAGC |
| | rev | GTCCCACGCTATAGTTGGAC |
| *off-target_OT-5* | fwd | CAGTATTCGTTATAACTTGGACC |
| | rev | GTCCTGCGCTGAAATTCGATC |
| *off-target_OT-6* | fwd | GACGAATATAAGGATGAGACATTG |
| | rev | GCGTACTATGGCTCGAGC |
| *off-target_OT-7* | fwd | AGCCGATGATGCAACCATC |
| | rev | CCAACTGTCTTTGTCCAGC |

All primers are listed by name, sequences and application. Forward primers are indicated as 'fwd' and reverse primers as 'rev'.

# Materials and methods

## Identification and cloning of *cpmrp*

The full-length cDNA corresponding to the predicted open reading frame of *cpmrp* (Gen-Bank accession number KC112554) was identified from our cDNA library of *C. populi* glands, qPCR validation and subsequent rapid amplification of cDNA ends (RACE) PCR using BD SMART RACE cDNA Amplification Kit (BD Biosciences, Heidelberg, Germany) according to the manufacturer's guidelines (see *Table 1* for primer sequences). After confirming the fidelity of the 4-kb amplification product by sequencing, it was cloned into pIB-V5-His-TOPO (Invitrogen). Amplification of *Cp*MRP homologs of *P. cochleariae* (*Pc*MRP; Gen-Bank accession number KF278996) and *C. lapponica* (*Clap*MRP; Gen-Bank accession number KF278997) were achieved by using the SMART RACE protocol and the primer *cpmrp* 5′-RACE.

## Sequence analysis and *Cp*MRP modeling

Sequence similarities were analyzed using the alignment tool BLAST (*Altschul et al., 1990*). Multiple alignment of MRP amino acid sequences was carried out with CLUSTALW using default parameters (DNASTAR Lasergene 10 Core Suite software, Madison, WI). The I-TASSER online server (*Roy et al., 2010*) was employed to predict a 3D structure model of *Cp*MRP based on its complete amino acid sequence. In a multistep modeling process, a total of about 20% of the sequence of P-glycoprotein from *Mus musculus* (*Aller et al., 2009*) and *Caenorhabditis elegans* (*Jin et al., 2012*) served as template fragments to initiate the structure modeling. Additionally, the *Cp*MRP structure model was embedded into a lipid bilayer using the CHARMM-GUI Membrane Builder (*Jo et al., 2009*). Typical sequence motifs of *Cp*MRP were identified by a sequence alignment using NCBI–Protein BLAST (*Altschul et al., 1990*). Finally, the program VMD (Visual Molecular Dynamics) was used to visualize the model (*Humphrey et al., 1996*).

## Rearing, maintaining and dissecting *Chrysomela populi*

A starting culture of roughly 100 *C. populi* (L.) larvae was collected near Dornburg, Germany (+51°00′52.00″, +11°38′17.00″), on *Populus maximowiczii x Populus nigra*. The larvae/beetles were kept for 5 months in a light/dark cycle of 16 hr light and 8 hr darkness (LD 16/8) at 18°C ± 2°C in light and 13°C ± 2°C in darkness. For RNAi experiments we used 3- to 4-day-old larvae of *C. populi* that were reared separately. DNA and RNA were isolated from larvae of *C. lapponica*, which were collected from *Betula rotundifolia* in the Altai Mountains, East Kazakhstan, (2130 m altitude, +49°07′4.38″, +86°01′3.65″) and from *P. cochleariae* (F.) larvae reared in a continuous lab culture (kept in a York Chamber at 15°C (LD 16/8) on leaves of *Brassica rapa pekinensis*). Larvae were dissected for tissue preparation in saline solution and directly frozen in liquid nitrogen.

## Collection of larval secretion

Larval secretions were collected and weighed in glass capillaries on an ultra-microbalance (Mettler-Toledo, Greifensee, Switzerland).

## RNAi in *C. populi* larvae

Sequence-verified plasmid pIB-*Cp*MRP was used to amplify a 730 bp fragment of *cpmrp* dsRNA. As a control, a *gfp* sequence was amplified from pcDNA3.1/CT- GFP-TOPO (Invitrogen). The amplicons were subject to an in vitro-transcription assays according to instructions from the Ambion MEGAscript RNAi kit (Life Technologies, Darmstadt, Germany; see *Table 1* for primer sequences). The resulting dsRNA was eluted after nuclease digestion three times with 50 µl of injection buffer (3.5 mM Tris-HCl, 1 mM NaCl, 50 nM $Na_2HPO_4$, 20 nM $KH_2PO_4$, 3 mM KCl, 0.3 mM EDTA, pH 7.0). The quality of dsRNA was checked by TBE-agarose-electrophoresis.

First-instar of *C. populi* (3–4 days after hatching) with 3–4 mm body length (chilled on ice) were injected with 0.25 µg of dsRNA by using a nanoliter microinjection system (WPI Nanoliter 2000 Injector, World Precision Instruments, Berlin, Germany). Injections were made into the hemolymph next to the ventral side between the pro- and mesothorax. Relative transcript abundance was quantified by quantitative real-time PCR (qPCR) at different stages of larval development after RNAi treatment. In silico off-target prediction was done for highly specific silencing according to (*Bodemann et al., 2012*). Experimentally, we excluded off-target effects based on the analysis of co-silencing-effects on non-target genes using qPCR, SDS-PAGE and Western blot. We chose seven of the ABC transporter sequences most similar to *Cp*MRP from our cDNA library; those shared 62–76% aa-sequence identity (*Figure 4—figure supplement 3*) were seen as best potential off-targets, so we analyzed

their transcript levels in *cpmrp* dsRNA injected- and *gfp*-injected control larvae. We found no off-target effect on the transcript level when qPCR was used in different larvae tissue (*Figure 4—figure supplement 4*; see *Table 1* for primer sequences). On protein level, we probed off-target effects via SDS-PAGE and Western blot comparing protein samples of *cpmrp* dsRNA injected- and *gfp*-injected control larvae (*Figure 4—figure supplement 1*).

## Quantitative real-time PCR (qPCR)

Total RNA was extracted from larval tissue using an RNeasy MINI kit (Qiagen, Hilden, Germany). cDNA was synthesized from DNA-digested RNA using RNAqueous micro kit (Life Technologies, Darmstadt, Germany). Realtime PCR was performed using Brilliant II SYBR Green qPCR Master Mix (Agilent) according to the manufacturer's instructions and an Mx3000P Real-Time PCR system. *CpActin* and *CpRPL45* expression was used to normalize transcript quantities for *C. populi*, *eIF4A* (*Kirsch et al., 2011*) for *C. lapponica* and *pcRP-L8* for *P. cochleariae* samples (see *Table 1* for primer sequences). Analyses were performed according to the MIQE-guidelines (*Bustin, 2010*; *Bustin et al., 2010*).

## *Xenopus laevis* oocytes isolation and RNA injection

*Cpmrp* RNA was generated by in vitro transcription using mMESSAGE mMACHINE kit (Ambion, Life Technologies, Darmstadt, Germany). *X. laevis* oocytes were provided by Prof Stefan H Heinemann (FSU Jena, Germany). 100–125 ng of cRNA was injected per oocyte (RNase-free water was used as a control). Oocytes expressing *Cp*MRP were maintained at 17.5°C in modified Barth's medium (MBS, in mM: 88 NaCl, 1 KCl, 2.4 NaHCO$_3$, 0.82 MgSO$_4$, 0.33 Ca(NO$_3$)$_2$, 0.41 CaC1$_2$, TRIS-HCL, pH 7.4) with 10 µg ml$^{-1}$ penicillin, 10 µg ml$^{-1}$ streptomycin and 4 µg ml$^{-1}$ cefuroxim solution for 3 days.

## Substrate efflux assay in *X. laevis* oocytes

Functional efflux studies were carried out with different glucosides in *X. laevis* oocytes at room temperature. The transport activity assay was initialized by injecting individual substrates or a substrate mixture (3 days post cRNA-injection). In comparative transport assays using a substrate mixture the control value was used to normalize the transport rates. The oocytes were immediately washed in Barth medium after substrate injection. At defined time points, the incubation medium was removed and analyzed either by HPLC-MS or UV detection at 268 nm to quantify the substrate efflux from the oocytes into the incubation medium. The calculation of the kinetic parameters was performed with GraphPad Prism (version 5.04, Graphpad Software. San Diego, CA) using the built-in enzyme kinetics module. The substrate concentration was based on the assumption of an oocyte volume of 1 µl (*Kelly et al., 1995*).

## HPLC-MS analysis

The efflux of the injected substrates was monitored in the oocyte incubation medium via HPLC-MS. An Agilent HP1100 HPLC system equipped to a C18 column (Gemini 5 µ C18 110A 250 × 2.00 mm 5 µm (Phenomenex, Aschaffenburg, Germany) was used for separation; analytes were detected APCI/MS (LCQ, Thermoquest, San Jose, CA) in positive mode. Samples were analyzed by using a gradient elution at 0.35 ml min$^{-1}$ (solvent A: H$_2$O+0.5% CHOOH; solvent B: MeCN+ 0.5% CHOOH) according to the following protocol: starting with 5% B, holding to 3 min, going to 20% in 12 min, going to 98% in 10 min, with subsequent washing. Peak areas from MS-chromatograms were obtained using an ICIS-algorithm (Xcalibur bundle version 2.0.7, Thermo Scientific, Waltham, MA).

## Statistical analyses

Two-tailed student's *t* tests for unequal variation were used to value significance levels.

## Live staining of glands

Dissected glands of *C. populi* were stained with the vital vacuolar stain 5-carboxy-2,7-dichlorofluorescein diacetate (CDCFDA). CDCFDA was added at 10 µM in the saline solution (with 50 mM sodium citrate, pH 5.0) for 20 min. The cells were co-stained with Hoechst 33342 and CellTrace BODIPY TR methylester (Image-iT LIVE Intracellular Membrane and Nuclear Labeling Kit, Invitrogen, Life Technologies, Darmstadt, Germany) for 10 min and immediately examined by two-photon imaging.

## Immunolabeling

Immunolabeling was employed to specifically localize *Cp*MRP in the defensive glands or whole larvae sections, respectively. For an overview staining, entire larvae of *C. populi* were used at

second-instar stage. The larvae were anesthetized in $CO_2$, directly embedded and shock-frozen in optical cutting temperature compound (OCT; Sakura Finetec, Staufen, Germany). 12–20 µm vibratome sections (Microm HM560, Thermo Scientific, Waltham, MA) were prepared.

Both dissected glandular tissue and fresh frozen sections were fixed in cold 4% PFA in 0.1 M PBS (pH 7.4) at 4°C for 1 or 2 hr, respectively. After permeabilization, the samples were washed 3 × 20 min in PBS-TX (0.5%), then blocked with NGS (normal goat serum) for 2 hr at room temperature (RT) and subsequently incubated in the primary antisera for 1 hr at RT and another 12 hr at 4°C. Polyclonal rabbit anti-*Cp*MRP sera (synthetic peptide antibody against the peptide mix: C+LKDVAEKAYHKNSRL [aa 1317–1331] and SLDGNKYTNENRDFS+C [aa 760–774]) were generated by Eurogentec (Seraing, Belgium) and used as primary antiserum at a concentration of 1:1000 in PBS-TX. After incubation in the primary antisera, the tissue was washed with PBS-TX (3 × 20 min) at RT and then incubated in secondary antisera conjugated to Alexa 488 (Invitrogen, Darmstadt, Germany) at a concentration of 1:500 in PBS, overnight at RT. Finally, the tissues were washed in PBS-TX (3 × 20 min) and PBS (20 min) at RT and mounted in Vectashield fluorescence mounting medium (Vector Labs, Peterborough, UK) in spacer slides (Grace Biolabs, Bend, Oregon). Hoechst 33342 was used to co-stain the nuclei, Bodipy to stain intracellular membranes following second antibody incubation, as described.

## Two-photon imaging

In order to analyze fluorescence in the glands, we employed an inverted multiphoton laser scanning microscope (Axio Observer Z.1 and LSM 710 NLO, Zeiss, Jena, Germany) in combination with a femtosecond Ti:Sapphire laser (80 MHz, 150 fs, Chameleon Ultra, Coherent Inc.) and a Plan-Apochromat 63x/1.4 objective (Zeiss, Jena Germany). An excitation wavelength of 800 nm (~2 mW) was used for glands stained with CDCFDA and 930 nm (~9 mW) in case of Alexa 488 staining. Emission wavelength detection was achieved by either integration over three spectral ranges (Hoechst 33342: 420–490 nm; Alexa 488/CDCFDA: 505–555 nm; Bodipy: 620–680 nm) or recorded spectrally resolved.

In order to separate autofluorescence from exogenous fluorophores all pixels of spectrally resolved images were assigned to individual fluorophores using a linear unmixing algorithm (Zen 2011 software; Zeiss, Jena Germany). ImageJ was used for image deconvolution (Diffraction PSF 3D; Iterative Deconvolve 3D) and 3D reconstruction (3D Viewer). *Cp*MRP degradation kinetics were estimated by integrated Alexa 488 fluorescence intensity from secretory cells normalized to autofluorescence. Spectrally resolved images were linear unmixed to separate Alexa 488 from autofluorescence. Autofluorescence images were corrected for all non-comparable contributions. The decay of Alexa 488 fluorescence was approximated by a monoexponential fit according to $N(t) = N_0 * \exp(-kt) + c$.

## Protein extraction and Western blot

Proteins were extracted from the dissected glandular larvae tissue by sonication in 10 mM Hepes, pH 7.5, 150 mM NaCl, 0.1% Triton, 1 mM dithiothreitol and protease inhibitor mix M (Serva, Heidelberg, Germany). Membrane protein fraction was roughly separated from the cytosolic fraction by centrifugation step at 20,000 × *g* at 4°C for 30 min. Proteins were separated by SDS-PAGE (Any kD Precast Gel, BioRad, Hercules, CA) and then transferred onto a PVDF membrane. The membrane was incubated first with rabbit anti-*Cp*MRP antibody (for details see 'Immunolabeling' section) and then with donkey anti-rabbit IgG-horseradish peroxidase conjugates (GE Healthcare Life Sciences, Freiburg, Germany). The proteins were detected using the SuperSignal West Dura Extended Duration Substrate Kit (Pierce Protein Natural Products, Thermo Fisher Scientific Inc. Rockford, IL).

## Acknowledgements

The authors would like to express their gratitude to Stefan H Heinemann for offering *Xenopus* oocytes and equipments, Jens Haueisen for providing microscopic equipment, Angelika Berg, Franziska Eberl, Sandra Klemmer, Angela Rossner, Regina Stieber, Roy Kirsch, Anja David and Maritta Kunert for technical assistance. We wish to thank Gergely Szakacs, David Heckel and Jacques M Pasteels for advice and helpful discussions on aspects of this work and Emily Wheeler for editorial assistance. This work has been supported by the Max Planck Society.

Dedicated to Prof Dr L Jaenicke on the occasion of his 90[th] birthday.

# Additional information

### Funding

| Funder | Grant reference number | Author |
|--------|------------------------|--------|
| Max Planck Society | | Anja S Strauss, Wilhelm Boland, Antje Burse |
| Deutsche Forschungsgesellschaft | BU1862/2-1 | Antje Burse |
| University Hospital Jena | | Sven Peters |

The funders had no role in study design, data collection and interpretation, or the decision to submit the work for publication.

### Author contributions

ASS, SP, AB, Conception and design, Acquisition of data, Analysis and interpretation of data, Drafting or revising the article; WB, Conception and design, Drafting or revising the article, Contributed unpublished essential data or reagents

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
