## [Decision Letter]

Thank you for sending your work entitled “Beetle juice strategy: ABC transporter functions as a pacemaker for sequestration of plant glucosides in leaf beetles” for consideration at *eLife*. Your article has been favorably evaluated by a Senior editor, Detlef Weigel, Marcel Dicke who is a member of our Board of Reviewing editors, and 2 peer reviewers.

You provide compelling evidence that *Cp*MRP is a low-affinity transporter involved in the sequestration of salicin by the defensive glands of poplar leaf beetle larvae. This careful study with a non-model organism adds new and substantial insight into the function of ABC transporters in insects and how insects deal with noxious compounds produced by plants they feed on. You demonstrate the key importance of this intracellular carrier for the defense mechanism of the beetle larvae by well-controlled RNAi experiments. These findings help to elucidate not only cell biological processes in the sequestration process, but also shed light on the evolution of diverse defense systems in leaf beetles.

The Reviewing editor and other reviewers discussed their comments before we reached this decision, and the Reviewing editor has assembled the following comments to help you prepare a revised submission. This is a very interesting study with many experimental data. Although the reviewers overall read the manuscript with interest, there are some changes that are needed.

1) Providing experimental evidence (Western blot) that the RNAi data are not a result of off-target effects.

2) The indication of concentrations when calculating the Km values.

3) Including comparative discussion considering possible *cpmrp* homologs in other insects.

4) Providing more information on the beetles used in the Materials and methods section, such as: origin of the beetles, rearing methods and the food used, number of beetles used to start the colony, and information on the *P. cochleariae* beetles that were used in the experiments.

5) The paper is not very clearly written and the lack of detail in the description of the results was a problem. Similarly, the authors need to use punctuation to clarify their meaning in many places. The English language throughout the text should be improved.

6) Finally the first three words of the title do not seem to connect to the contents of the manuscript and distract from the message of the manuscript and its title.

---

## [Author Response]

*1) Providing experimental evidence (Western blot) that the RNAi data are not a result of off-target effects*.

Experimentally, we excluded off-target effects based on the analysis of co-silencing-effects on non-target genes using qPCR, SDS-Page and Western blot. We have included Figure 3—figure supplement 1: “Effects of *cpmrp* silencing on the protein level of the glandular tissue of *C. populi*” to provide additional experimental evidence that the RNAi data are not a result of off-target effects.

*2) The indication of concentrations when calculating the Km values*.

We have revised Figure 4 and indicated the injected salicin in relation to the oocyte volume to calculate substrate concentrations.

*3) Including comparative discussion considering possible* cpmrp *homologs in other insects*.

We agree that a comparable discussion of *cprmp* homologs could extend our sequestration model to other insects but it is rather limited and speculative to date. In insect databases there are predicted ABC transporter sequences homologous to *Cp*MRP identified in *Chrysomela populi* (the highest amino acid sequence similarity (75%) with 59% identity was found to the predicted ATP-binding cassette transporter of *Tribolium castaneum* (TC014776)). However, these sequences have been neither functionally characterized nor localized in the insects to date. Based on the uniform morphology of insect epidermal glands only, we would speculate the presence of *cpmrp* homologs in class 3 gland cells of insects (e.g., dermal glands of *Tenebrio molitor*, sternal glands of *Periplaneta americana*, pheromone gland of *Harpobittacus australis (*Noirot and Quennedy, 1974)). This need to be further investigated. In our manuscript we therefore focused on the discussion of the proposed sequestration model including *Cp*MRP homologs of other leaf beetles (*C. lapponica and P. cochleariae*) and their general relevance in exocrine glands of Chrysomelina species.

*4) Providing more information on the beetles used in the Materials and methods section, such as: origin of the beetles, rearing methods and the food used, number of beetles used to start the colony, and information on the* P. cochleariae *beetles that were used in the experiments*.

We have provided all information for collection and rearing of the leaf beetle species used in the study.

*5) The paper is not very clearly written and the lack of detail in the description of the results was a problem. Similarly, the authors need to use punctuation to clarify their meaning in many places. The English language throughout the text should be improved*.

We have edited the Results and Discussion parts. Particularly, the last paragraphs of this section were completely revised. We have included a conclusion in order to provide a clear take-home message for a broad readership. We have corrected typos and we have improved the English language throughout the manuscript.

*6) Finally the first three words of the title do not seem to connect to the contents of the manuscript and distract from the message of the manuscript and its title*.

We have changed the title to read: “ABC transporter functions as a pacemaker for the sequestration of plant glucosides in leaf beetles”